# Assessment of the Smartpill, a Wireless Sensor, as a Measurement Tool for Intra-Abdominal Pressure (IAP)

**DOI:** 10.3390/s24010054

**Published:** 2023-12-21

**Authors:** Andréa Soucasse, Arthur Jourdan, Lauriane Edin, Elise Meunier, Thierry Bege, Catherine Masson

**Affiliations:** 1IFSTTAR, Université Gustave Eiffel, Aix-Marseille Université, Laboratoire de Biomécanique Appliquée, Faculté de Médecine, Campus Nord, Boulevard Pierre Dramard, CEDEX 20, 13916 Marseille, France; 2Assistance Publique des Hôpitaux de Marseille (APHM), Aix-Marseille Université, Hôpital Nord, Service de Gastro-Entérologie, Chemin des Bourrely, 13015 Marseille, France; 3Assistance Publique des Hôpitaux de Marseille (APHM), Aix-Marseille Université, Hôpital Nord, Service de Chirurgie Générale, Chemin des Bourrely, 13015 Marseille, France

**Keywords:** abdominal wall, mechanical behavior, ingestible capsule, intra-abdominal pressure, intra-gastric pressure, porcine model

## Abstract

***Background:*** The SmartPill, a multisensor ingestible capsule, is marketed for intestinal motility disorders. It includes a pressure sensor, which could be used to study intra-abdominal pressure (IAP) variations. However, the validation data are lacking for this use. ***Material and Methods:*** An experimental study was conducted on anesthetized pigs with stepwise variations of IAP (from 0 to 15 mmHg by 3 mmHg steps) generated by laparoscopic insufflation. A SmartPill, inserted by endoscopy, provided intragastric pressure data. These data were compensated to take into account the intrabdominal temperature. They were compared to the pressure recorded by intragastric (IG) and intraperitoneal (IP) wired sensors by statistical Spearman and Bland–Altmann analysis. ***Results:*** More than 4500 pressure values for each sensor were generated on two animals. The IG pressure values obtained with the SmartPill were correlated with the IG pressure values obtained with the wired sensor (respectively, Spearman ρ coefficients 0.90 ± 0.08 and 0.72 ± 0.25; bias of −28 ± −0.3 mmHg and −29.2 ± 0.5 mmHg for pigs 1 and 2). The intragastric SmartPill values were also correlated with the IAP measured intra-peritoneally (respectively, Spearman ρ coefficients 0.49 ± 0.18 and 0.57 ± 0.30; bias of −29 ± 1 mmHg and −31 ± 0.7 mmHg for pigs 1 and 2). ***Conclusions:*** The SmartPill is a wireless and painless sensor that appears to correctly monitor IAP variations.

## 1. Introduction

Intra-abdominal pressure (IAP) is the ambient pressure within the abdominal cavity. It is routinely measured in intensive care units for the diagnosis and monitoring of intra-abdominal hypertension (IAH) and the associated abdominal compartment syndrome [1]. The physiological consequences of IAH and its negative impact on morbidity and mortality are well known, leading to abdominal decompression surgery [2,3]. Patient-specific assessment of IAP could also be interesting for abdominal surgery. Indeed, it would make it possible to know the stresses applied to the patient’s abdominal wall and thus guide the choice of surgical approach (implant, fixation) and predict the risk of recurrence as well as the early detection of complications [4].

Numerous studies have focused on the exploration of IAP over the past thirty years. The only currently validated techniques are based on direct and indirect methods with measurements performed in the bladder, stomach, or rectum. They are mostly invasive, inconvenient, and have the risk of infection.

The current reference measurement of IAP in clinical practice is the measurement of intra-vesical pressure. This is an indirect intermittent measurement technique, defined by Kron and validated by Iberti and then Fusco [5,6,7], for patients in the supine position, requiring a urinary catheter. This method is invasive, non-continuous, costly in medical or paramedical time, and it presents infectious risks. Kron’s method’s precision is limited because it is practitioner-dependent. Based on the measurement of the height of a water column, the zero level corresponding to the “mid-height” of the pubis is made at each measurement by the practitioner. Other validated alternative methods exist, such as intra-gastric pressure measurement [8,9] or intra-rectal pressure measurement [10], but they have the same drawbacks. All three measurement locations are based on the assumption of equal pressure within the abdominal cavity and the consideration of the abdominal contents as incompressible fluid [11,12,13].

Minimally invasive methods, which would allow for more systematic practice in the clinic, have been developed more recently. They are based on technologies such as bioimpedance, tensiometry, laser ultrasound, respiratory inductance plethysmography, a wireless intra-vaginal sensor, or an ingestible intra-gastric sensor, to name a few [14,15]. However, none of these technologies have yet been validated in clinical practice for the measurement of IAP.

Among these innovative techniques, the SmartPill™ (Medtronic, Minneapolis, MN, USA) seems promising [16,17]. It is an ingestible capsule containing, among other things, a pressure sensor, whose location in the gastrointestinal tract can be determined using a pH sensor. It allows the measurement of intra-gastric pressure continuously by remote transmission to an external receiver box (SmartPill™ Motility Recorder, Medtronic, Minneapolis, MN, USA) without a painful catheter and without presenting any infectious risk.

The SmartPill™(SP) can be useful to evaluate IAP variations during physical standardized exercises or during daily life outside the controlled conditions of a laboratory [18]. However, the only validation study of the SmartPill™ as a tool for measuring IAP was conducted in a porcine model and concluded that IAP was underestimated compared to intra-vesical measurement [19]. Discontinuous measurements and no measurement of intra-peritoneal pressure are limitations of this study, and they narrow the scope of the system validation. A complementary study appeared necessary to improve it. Moreover, several capsule data export modes can be chosen, with a large variation in the results for the same pressure variation. The choice of the export mode is thus an important point for system validation. 

The aim of this work is to evaluate the SP as a tool for the measurement of IAP. To this end, a study on an anesthetized pig model subjected to IAP variations by intra-peritoneal CO_2_ insufflation was conducted, comparing the continuous measurement of intra-gastric pressure obtained by SP to intra-gastric and intra-peritoneal measurements obtained by referent sensors.

## 2. Material and Methods

The present protocol was approved by the referring animal ethics committee (APAFiS authorization n°28496). The rules of animal ethics were fully respected, in particular by the application of the 3R rule [20]. The study was performed in an experimental operating room authorized for animal experimentation (Surgical Study and Research Center (CERC), Aix-Marseille University).

### 2.1. Animals and Protocol Preparation

Two female feeder pigs (average weight 35 kg and minimum age of 3 months) were placed supine on a veterinary surgical table and pre-anesthetized with ketamine hydrochloride (10 to 15 mg/kg) and azaperone (0.1 mg/kg). Continuous anesthesia was then performed via electric syringe with Propofol (0.5 to 0.8 mg/kg/h) and Sufentanil (6 to 10 µg/kg/h) as analgesic. 

A venous line was placed on the atrial vein, and orotracheal intubation was performed. The animals were mechanically ventilated, and hemodynamic parameters (heart rate and SaO_2_) were continuously monitored.

An open laparoscopy was performed at the level of the umbilicus, and two laparoscopic trocars were positioned: a median periumbilical trocar (applied medical 10 mm balloon trocar) and a right lateral trocar (applied medical 5 mm). The periumbilical trocar was connected to a CO_2_ laparoscopic insufflator (Aesculap), allowing the pressure to vary in the abdomen in a controlled manner from 0 to 15 mmHg. 

Animals were euthanized at the end of the procedure after the deepening of general anesthesia by injection of 150 mg/kg of Pentobarbital (Dolethal).

### 2.2. Pressure Measurements

#### 2.2.1. Wired Sensors 

The reference measurement of IAP was obtained by a wired sensor in the intra-peritoneal position (WIP) placed through the right lateral trocar. The reference measurement of intra-gastric pressure was obtained by positioning a wired sensor in the stomach (WIG).

The WIP and WIG sensors were wired sensors (catheter pressure transducer MPR 500 Millar) connected to a control box (PCU-2000 Millar). These sensors allow the measurement of absolute pressure and can be used in air and in liquids. Their accuracy is ±1.5 mmHg, and their measuring range extends from −50 to +300 mmHg. The WIP and WIG were connected to a Slice Nano acquisition box (18-channel DTS), allowing simultaneous recording at a frequency of 20 Hz.

The WIG was placed thanks to a video return endoscope (Olympus, Evis Exera II CLV-180). The WIP was lowered through the laparoscopic side trocar sleeve, protruding into the peritoneal cavity by 5 cm. The positioning of the sensor tip was checked by laparoscopic vision from the medial trocar; the sensor cable was secured to the trocar by adhesive tape.

#### 2.2.2. SmartPill™

In parallel, a wireless intra-gastric pressure measurement was performed by the SP. The SP is an FDA-approved and CE-marked ingestible capsule. It has been marketed since 2006 by Medtronic™ for the evaluation of transit and functional digestive disorders in humans. Its dimensions are 26 mm long and 13 mm in diameter and it contains pH, temperature and pressure sensors (Figure 1). Its accuracy is ±0.28 for pH measurement, ±0.5 °C for temperature measurement, and ±3.6 mmHg for pressure measurement. The battery life of the device is more than 5 days.

The SP was calibrated at atmospheric pressure, room temperature, and pH using a buffer solution (pH 6), according to the recommendations provided by the manufacturer. It was then paired with a receiver box, allowing the recording of data transmitted by radio frequency at 2 Hz. The SP was placed inside the stomach with a video return endoscope, and the receiver was placed near the animal’s abdomen.

The experimental setup with the positioning of the different sensors is shown in Figure 2.

This figure represents the experimental setup, which is implemented on the porcine model. The positions of the wired sensors and the wireless sensor, the SmartPill™, are represented, as well as those of their respective receivers.

### 2.3. Pressure Variations

Once the sensors were positioned, a stabilization period (or rest period) was observed for 5 min. Then, the pressure in the intra-peritoneal cavity was increased from 0 to 15 mmHg in steps of 3 mmHg by insufflation of CO_2_. Then, the pressure was decreased following the same steps until 0 mmHg. A stabilization period of 2 min was respected at each step.

### 2.4. Data Processing

SP data were exported to a PC via MotiliGI™ Software (Medtronic, Minneapolis, MN, USA) provided by the manufacturer. The exported data were the “temperature compensated (TC)” data, defined as raw data (in mV) multiplied by the scale factor of the sensor and then adjusted for the calibration point (pH 6, atmospheric pressure, room temperature) and sensor deflection due to temperature variations.

In order to study the correlations between SP, WIG, and WIP, the WIP and WIG signals were subsampled to 2 Hz. The analysis and processing of the data were performed on MATLAB^®^.

### 2.5. Statistical Analysis

Correlations between SP and WIG and WIP were evaluated by Spearman’s ρ coefficient for each pressure step. The Bland and Altmann plot was used to evaluate the bias and limits of agreement (interval containing 95% of differences) between SP and WIG and SP and WIP for each pressure step [22,23].

## 3. Results

### 3.1. Pattern of the Different Signals

The SP, WIG, and WIP pressure versus time curves recorded in both animals are presented in Figure 3. This figure shows the different pressure steps during the increasing and decreasing phases. The pressure patterns, according to the different steps, seem similar between the sensors. An offset of approximately −30 mmHg can be observed between the SP and the wired sensors. The rest period was longer for pig 2, during which high pressure oscillations were recorded by the SP. The total number of measurements per sensor was 2641 for the first animal and 1895 for the second.

This figure shows the intra-gastric pressure (by the SmartPill™ in turquoise, by the wired sensor in light blue—in mmHg) and the intra-peritoneal pressure (by the wired sensor in dark blue—in mmHg) as a function of pressure steps (3 mmHg increase and decrease) imposed by laparoscopic insufflation for both animals.

### 3.2. Correlation between Intra-Gastric Measurements by SmartPill™ (SP) and Wired Sensor (WIG)

The correlation lines between the pressure measured by SP and the pressure measured by WIG for both pigs at each step are shown in Figure 4.

This figure shows the point clouds and the correlation lines between the pressure measured by the SmartPill™ (SP) (in mmHg) and the pressure measured by the wired intra-gastric sensor (WIG) (in mmHg) for both pigs at each pressure step.

The Spearman ρ coefficients for each step are shown in Table 1.

For all the measurement data, the relation between the SP and wired intra-gastric measurement for pig 1 was y = 1.05x − 28.0 and for pig 2: y = 1.03x − 29.3.

For pig 1, the average Spearman’s coefficient of all steps combined was ρ = 0.90 ± 0.08 (min = 0.67 and max = 0.96). 

For pig 2, the average Spearman coefficient of all steps combined was ρ = 0.72 ± 0.25. The observed correlations ranged from ρ = 0.61 to ρ = 0.94 for all steps except for the 0 and 3 mmHg steps during the increase phase and the 0 mmHg step during the decrease phase, for which the observed correlations were lower (min = 0.31 and max = 0.42).

The Bland and Altmann plots between SP and WIG pressure for the two pigs are shown in Figure 5, with the bias and limits of agreement values for each step presented in Table 1.

Bland and Altmann plots are represented for the SmartPill™ (SP) and wired intra-gastric sensor (WIG) for both animals.

In pig 1, for the entire data set, the Bland and Altmann plot showed a mean bias of −28.0 ± 0.3 mmHg with mean limits of agreement of ±0.4 mmHg for all steps.

In pig 2, a mean bias of −29.2 ± 0.5 mmHg was calculated with mean limits of agreement of ±1.1 mmHg for all steps. The steps for which the limits of agreement are the widest are the 0 (±4.5 mmHg) and 3 mmHg (±1.4 mmHg) steps in the increase phase and the 0 mmHg (±1.8 mmHg) step in the decrease phase. For all other steps, the limits of agreement do not exceed ± 0.8 mmHg.

### 3.3. Correlation between Intra-Gastric Measurements by SmartPill™ (SP) and Intra-Peritoneal Measurements (WIP)

The correlation between the pressure measured by SP and the pressure measured by WIP for both pigs is shown in Figure 6, and ρ coefficients as well as bias values for each step are shown in Table 1.

This figure shows the point clouds and the correlation lines between the pressure measured by the intra-gastric SmartPill™ (SP) (in mmHg) and the pressure measured by the wired intra-peritoneal sensor (WIP) (in mmHg) for both pigs at each pressure step.

For pig 1, the average Spearman’s coefficient of all steps combined was ρ = 0.49 ± 0.18. Correlations observed for pig 1 were higher than 0.55 for 12 mmHg and 15 mmHg steps during the increase phase and for all steps during the decrease phase until 3 mmHg (max = 0.70). 

For pig 2, the average Spearman coefficient of all steps combined was ρ = 0.57 ± 0.30. The correlations were better during the decrease phase (ranged from 0.81 to 0.93 except for the 0 mmHg step) than during the increase phase (ranged from 0.24 to 0.87).

Figure 7 shows the Bland and Altmann plots between SP and WIP pressure for the two pigs.

Bland and Altmann plots are represented for the SmartPill™ (SP) and wired intra-peritoneal sensor (WIP) for both animals.

In pig 1, for the entire data set, the Bland and Altmann plot showed a mean bias of −29.0 ± 1.0 mmHg with mean limits of agreement of ±1.8 mmHg for all steps.

In pig 2, a mean bias of −31.0 ± 0.7 mmHg was calculated with mean limits of agreement of ±1.8 mmHg for all steps. 

For the two pigs, the 0 mmHg step during the increase phase is the one for which the limits of agreement are the widest. They were ±3.6 mmHg in pig 1 and ±5.0 mmHg in pig 2.

## 4. Discussion

This study reports intra-gastric pressure measurements assessed by the commercial ingestible SP sensor and intra-gastric and intra-peritoneal wired sensor pressure measurements. Measurements were conducted in two anesthetized swine at different pressure steps (between 0 and 15 mmHg) generated by a laparoscopic CO_2_ insufflator. The results presented in this work are based on a large number of continuous measurements (on average, 2250 for each animal).

The results showed that the SP correctly reflects intra-gastric pressure changes for IAP greater than 3 mmHg (ρ = 0.92 ± 0.04 for pig 1 and ρ = 0.85 ± 0.12 for pig 2). The reliability of the SP intra-gastric measurement is also confirmed by low agreement limits that averaged ±0.4 mmHg in pig 1 and ±1.1 mmHg in pig 2 for all steps.

The study also showed that intra-peritoneal pressure changes are correctly reflected by SP for IAP greater than 3 mmHg (ρ = 0.57 ± 0.11 for pig 1 and ρ = 0.67 ± 0.27 for pig 2), although the correlations are lower than those obtained between SP and WIG. Again, the reliability of the measurement is evidenced by the low agreement limits that averaged ±1.8 mmHg for both animals. These results are in accordance with previous studies reporting equal pressure at any point in the abdominal cavity in the supine position [11,13].

For pressure levels between 0 and 3 mmHg, there is less reliability in the measurement of intra-gastric and intra-peritoneal pressure by SP. However, in the physiological situation of humans, the IAP is higher than this threshold, since it is between 5 and 7 mmHg at rest and can rise to values of over 100 mmHg depending on the exercise performed [24,25].

A measurement bias is observed between the SP-wired sensors, varying between −28 and −31 mmHg for both animals (shown in Figure 3). This bias is certainly related to the temperature increase during the placement of the SP in the animals’ stomachs. Indeed, before “ingestion”, the SP pressure was close to 0 mmHg. Once the capsule is in the stomach, the pressure drops, whereas the pressure measured by the wired sensors remains around 0 mmHg. This bias between SP and the wired sensors is then globally constant for the overall duration of the experiment. This phenomenon is illustrated in Figure 8. This bias prevents the measurement of the true pressure in the body, and only the pressure variations are quantifiable. A future collaboration with the industrial developer of this sensor would help further this point.

To our knowledge, only one validation study conducted by Rauch et al. evaluated SP for the measurement of IAP [19]. Their protocol differed significantly from the one presented here. Intra-gastric pressure measured by SP was compared with intra-vesical pressure in eight anesthetized pigs over a 24 h period without imposed pressure changes. Rauch et al. concluded that the pressure recorded by the SP was underestimated compared to the intra-vesical pressure. SP pressure ranged from 1 to 3 mmHg, whereas intra-vesical pressure ranged from 3 to 15 mmHg, resulting in a mean difference between the two measurements of 6.2 (±1.4) mmHg. They calculated limits of agreement ranging between ±3.3 and ±8.9 mmHg. Their limits are well above ours, which ranged between ±0.4 and ±1.8 mmHg. The differences in the results of Rauch et al. and ours could be explained by the choice of SP data export mode. In our work, we used the manufacturer’s proposed “temperature compensated” (TC) data export mode, whereas Rauch et al. appear to have used the default export mode called “baseline compensated” (BC). The BC mode corresponds to the TC data with a correction (not detailed by the manufacturer) when the measured pressure is negative. In Figure 9, the WIG and SP signals are compared according to the TC (with bias correction) and BC export modes. The BC signal is not at all correlated with the real pressure variations during the steps. Therefore, it is very important to point out that the BC export mode should be avoided for IAP variation measurements.

Intra-gastric pressure (measured by a wired intra-gastric sensor) and SmartPill™ (SP) pressure according to the “temperature-compensated” (TC) and “baseline-compensated” (BC) export modes; the bias of the TC signal was corrected by adding the mean bias measured during the test. 

The main limitation of our work stands in the limited number of animals. Nevertheless, in the present protocol and for this preliminary study purpose, we applied the 3R Rule [20]. Knowing the fact that we planned to record a large number of continuous measurements (more than 2000 per animal), we stated that two animals would be enough to have statistically significant results.

Another limitation of this study is the maximum IAP that was reached in our protocol. It would have been interesting to exceed 15 mmHg, but it was not possible due to the technical limitations of the laparoscopic column. However, as shown in Table 1, the correlation coefficients between SP and wired sensors improved with increasing pressure levels. This suggests that SP will accurately reflect IAP at higher pressure levels. In addition, a longer monitoring period for each animal would have been helpful to reinforce these correlations. 

Today, the ventral hernia repair recurrence rate remains high, reaching 28% after two years [26,27]. New diagnostic tools are needed to improve outcomes, and the assessment of the pressure profile with the SP could be an interesting tool for the patient’s management and the personalization of the surgical approach. Pressure variations during standardized exercises and during activities of daily life have so far been very little studied due to the lack of a suitable sensor. Yet, it is likely that this parameter strongly influences the risk of recurrence, as IAP is the main mechanical stress applied to the musculoaponeurotic wall and consequently to the scar zone. The present study reinforces the findings of a previous study, in which our team used the SP to build the pressure profile of 20 healthy volunteers [18]. 

However, in the context of the monitoring of intra-abdominal hypertension and abdominal compartment syndrome, the use of SP as a substitute for the measurement of intra-vesical pressure does not appear to be suitable. The patients concerned by these pathologies are not good candidates for ingesting a capsule because they frequently present intestinal occlusion or ileus.

## Figures and Tables

**Figure 1 sensors-24-00054-f001:**
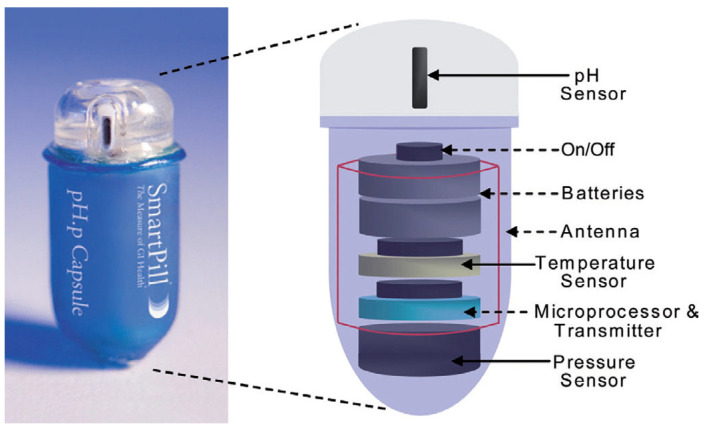
Image and schematic of the components of the SmartPill^®^; reprinted from https://doi.org/10.1016/S1369-7021(09)70272-X (accessed on 2 December 2023) [21].

**Figure 2 sensors-24-00054-f002:**
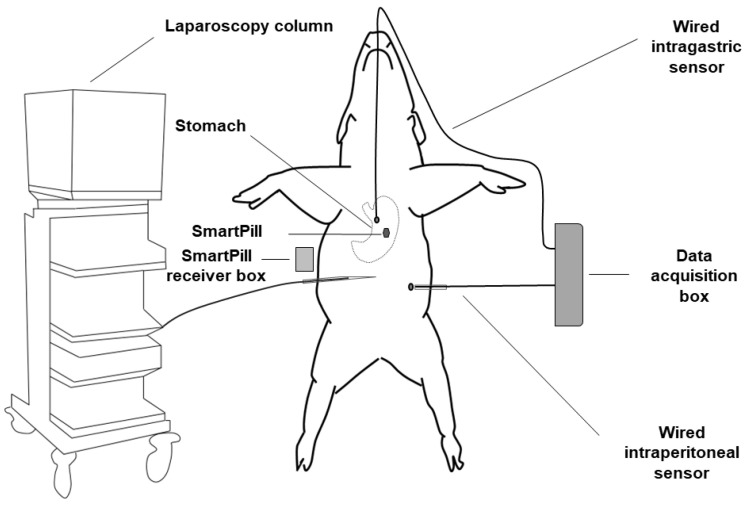
Schema of the experimental setup.

**Figure 3 sensors-24-00054-f003:**
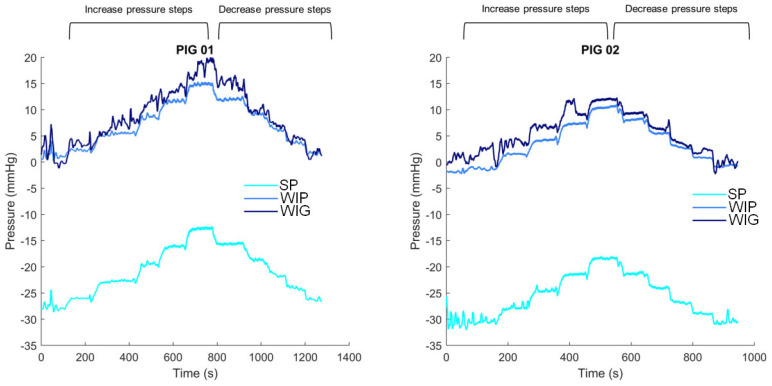
Intra-gastric and intra-peritoneal pressure as functions of pressure steps—patterns of the different signals.

**Figure 4 sensors-24-00054-f004:**
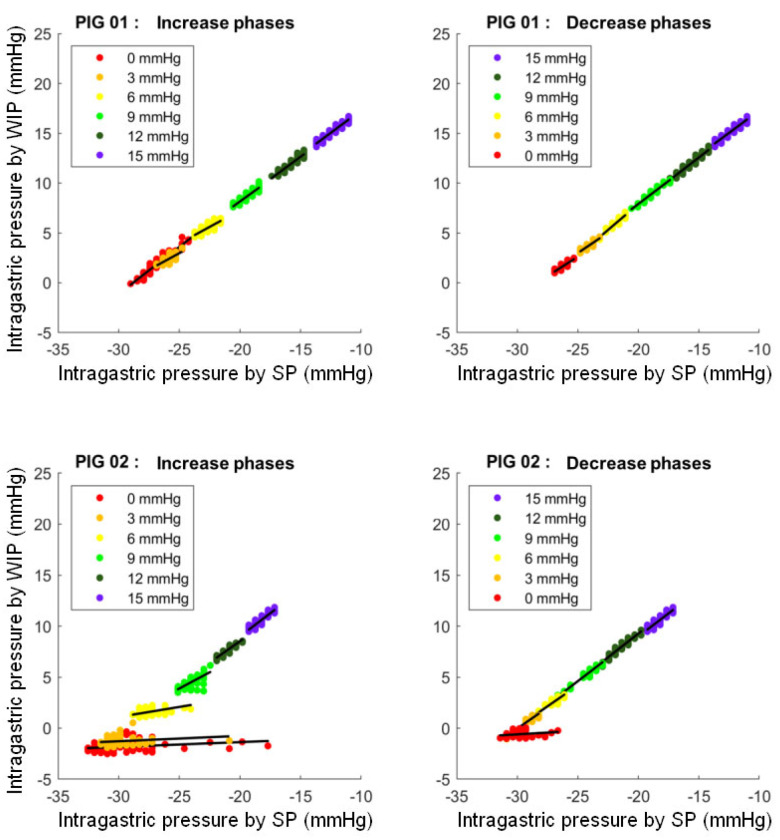
Correlation between intra-gastric pressure measured by the SmartPill™ and the wired sensor.

**Figure 5 sensors-24-00054-f005:**
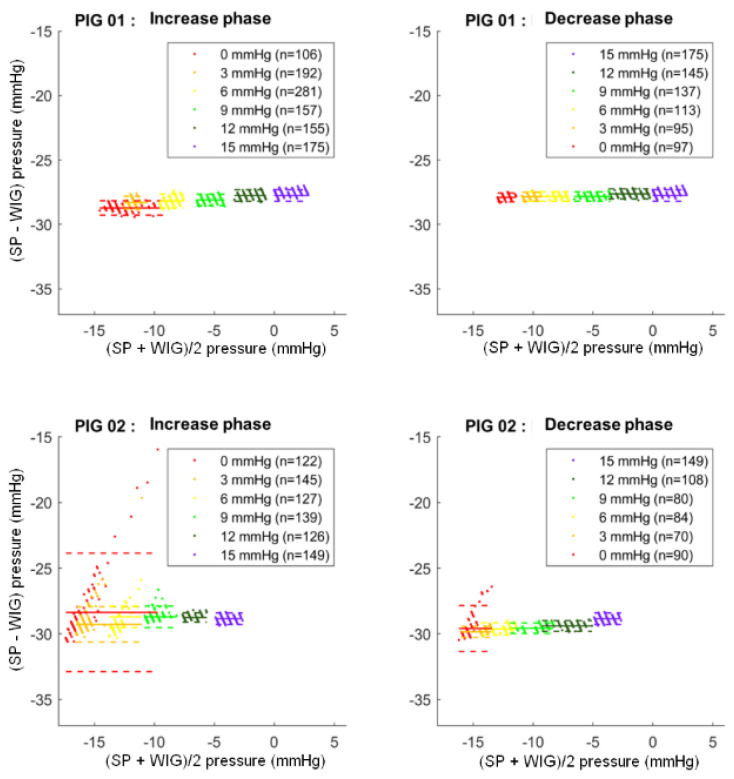
Bias of measurement between intra-gastric pressure measured by the SmartPill™ and intra-gastric pressure measured by wired sensor.

**Figure 6 sensors-24-00054-f006:**
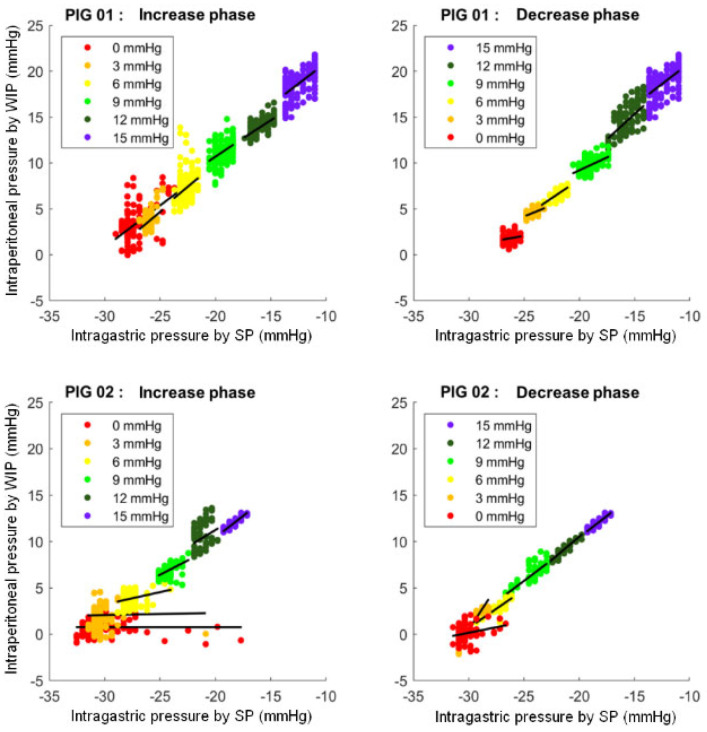
Correlation between pressure measured by the SmartPill™ and the wired intra-peritoneal sensor.

**Figure 7 sensors-24-00054-f007:**
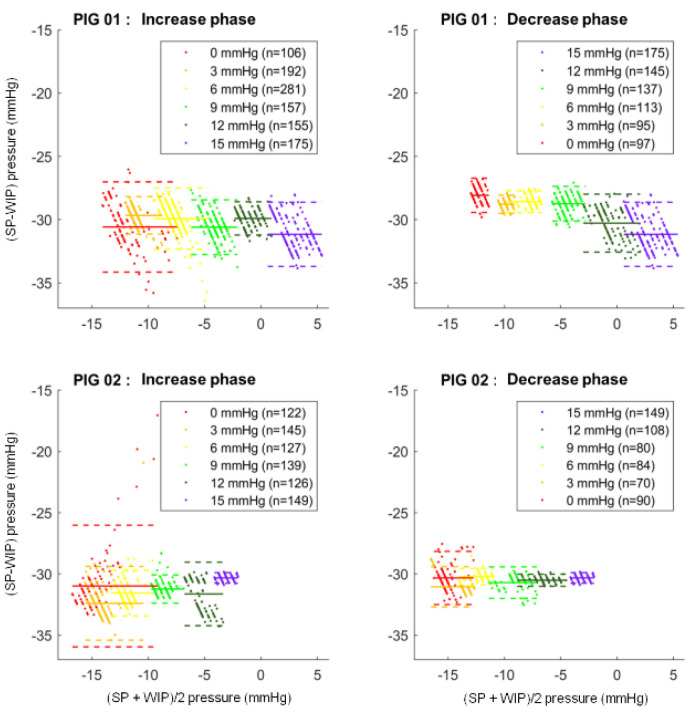
Bias of measurement between intra-gastric pressure measured by the SmartPill™ and intra-peritoneal pressure measured by a wired sensor.

**Figure 8 sensors-24-00054-f008:**
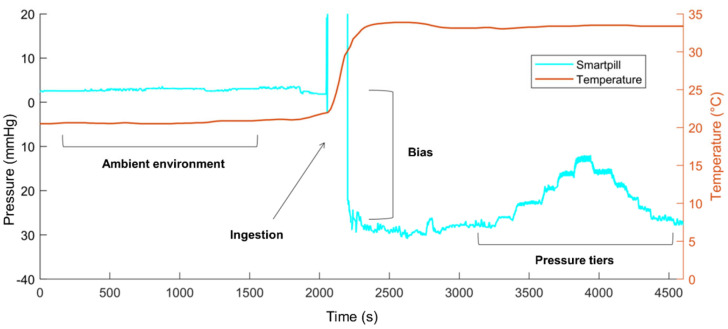
Illustration of the SmartPill™ measurement bias. This figure represents the bias of measurement that appears of the pressure (mmHg) versus time (s) curve when placing the SmartPill™ (SP) capsule in the pig’s 1 stomach. Curve of the temperature (°C) as a function of time (s) is also represented.

**Figure 9 sensors-24-00054-f009:**
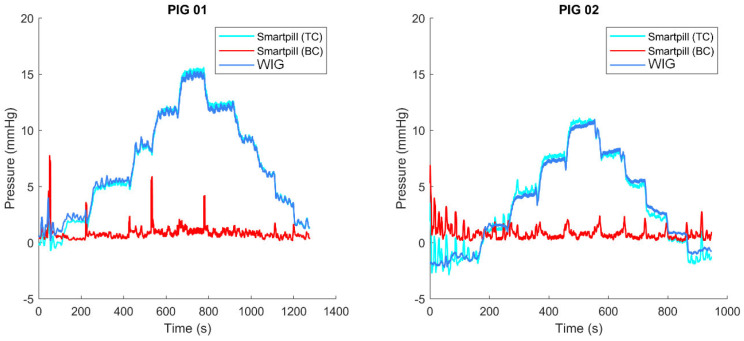
Comparison of the pressure measured by the SmartPill™ depending on chosen data export mode.

**Table 1 sensors-24-00054-t001:** Spearman’s ρ coefficients, measurement bias and limits of agreement between wired intra-gastric sensor (WIG) and SmartPill™ (SP) and between wired intra-peritoneal sensor (WIP) and SmartPill™ (SP) for all pressure steps.

		PIG 1						PIG 2					
		WIG vs. SP			WIP vs. SP			WIG vs. SP			WIP vs. SP		
Steps (mmHg)		ρ	Bias	Limit	ρ	Bias	Limit	ρ	Bias	Limit	ρ	Bias	Limit
			(mmHg)			(mmHg)			(mmHg)			(mmHg)	
Increase	0	0.94	−28.7	0.6	0.34	−30.6	3.6	0.42	−28.4	4.5	0.28	−31	5.0
	3	0.67	−28.3	0.5	0.31	−29.7	1.5	0.31	−29.3	1.3	0.23	−32.4	3.0
	6	0.83	−28.2	0.5	0.46	−29.9	2.4	0.61	−28.7	0.8	0.31	−31.5	1.9
	9	0.94	−28.1	0.4	0.39	−30.6	2.2	0.79	−28.7	0.8	0.58	−31.2	1.2
	12	0.93	−27.7	0.4	0.70	−29.9	1.3	0.90	−28.7	0.4	0.3	−31.6	2.6
	15	0.93	−27.6	0.4	0.58	−31.2	2.5	0.87	−28.8	0.4	0.87	−30.3	0.4
Decrease	12	0.96	−27.6	0.4	0.64	−30.3	2.3	0.94	−29.4	0.4	0.93	−30.5	0.5
	9	0.96	−27.8	0.4	0.55	−28.7	1.4	0.94	−29.6	0.4	0.81	−30.7	1.3
	6	0.92	−27.8	0.4	0.69	−28.6	0.9	0.92	−29.6	0.5	0.89	−30.2	0.7
	3	0.92	−27.8	0.3	0.57	−28.8	0.7	0.89	−29.8	0.4	0.86	−31	1.6
	0	0.92	−27.9	0.3	0.13	−28.1	1.4	0.37	−29.6	1.7	0.22	−30.3	2.2
Mean		0.90	−28.0	0.4	0.49	−29.7	1.8	0.72	−29.2	1.1	0.57	−31.0	1.8
SD		0.08	0.3	0.1	0.18	1.0	0.9	0.25	0.5	1.2	0.30	0.7	1.3

## Data Availability

Data are contained within the article.

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
