# Peer review of "Assessment of the Smartpill, a Wireless Sensor, as a Measurement Tool for Intra-Abdominal Pressure (IAP)"

_sensors, 2023, doi:10.3390/s24010054_

Round 1

Reviewer 1 Report

Comments and Suggestions for Authors

There is high need to measure easily intraabdominal pressure. The paper address this issue by assessing performance of commercial Smartpill. The topic is worth studying and the results are important. Below I have some small comments:

1. Could you comment in the manuscript on the results presented in your previous paper” A better understanding of daily life abdominal wall mechanical solicitation: Investigation of intra-abdominal pressure variations by intragastric wireless sensor in humans”. Are these results valid in the view of your current paper?

2. Why the sensor is calibrated in the room temperature?  Would it be possible to calibrate it in the temperature in which it will work? Is relation between temperature  and sensor deflection well established for the needed range of temperatures?

3. Is the relation between wire measurement and Smartpill similar between two subjected (pigs).

 4. There are some language typos in the text.

Reviewer 2 Report

Comments and Suggestions for Authors

The paper reports the works using SmartPill to measure the intra-abdominal pressure (IAP) variations. The results were compared to the pressures recorded by intragastric (IG) and intraperitoneal (IP) wired sensors by statistical Spearman and Bland Altmann analysis. teh results show that the SmartPill is a wireless and painless sensor that appears to correctly monitor IAP variations. The paper is completed in the present form. The reviewer recommends publishing it in the journal.

Reviewer 3 Report

Comments and Suggestions for Authors

The manuscript presents a study on the use of the SmartPill ingestible capsule sensor for monitoring intra-abdominal pressure (IAP) variations in anesthetized pigs. The study primarily focuses on validating the sensor's performance in measuring IAP and its potential clinical applications. The key innovation lies in the application of the SmartPill sensor for continuous IAP monitoring. The study innovatively explores the application of the SmartPill sensor for continuous intra-abdominal pressure monitoring, providing a potential non-invasive alternative to traditional methods. The continuous monitoring aspect of the study is noteworthy, allowing for a more comprehensive understanding of IAP dynamics. However, there are also some problems before publication.

1.  While the study provides valuable insights, there are methodological limitations that need to be addressed. The use of a small number of animals may limit the generalizability of the findings. Increasing the sample size or diversifying the animal models used could strengthen the study's robustness.

2.  The study primarily serves as a supportive validation of the commercial sensor's utility for IAP measurements. However, it appears to lack a substantial degree of novelty beyond confirming the sensor's capabilities. It would be beneficial to enhance the manuscript's innovation by delving deeper into potential different modes of measurement application or exploring novel clinical scenarios where the SmartPill sensor could make a significant impact.

In the revision, it is recommended to expand on the potential applications of the sensor in different measurement modes, thereby enriching the innovative aspects of the study. Additionally, addressing the methodological limitations and considering a broader range of animal models or clinical scenarios would enhance the manuscript's overall impact. Please make the necessary revisions to address these concerns and improve the manuscript's overall quality.

Comments on the Quality of English Language

The language of the manuscript is generally clear and fluent, with no apparent spelling errors or grammar issues. The authors use professional scientific terminology and provide detailed information when describing the research methods and results, which aids in comprehension of the study's processes and findings. However, in the discussion section of the article, I noticed that some sentences might require clearer and more detailed expression to ensure readers fully understand the authors' viewpoints and the significance of the research. Additionally, there are some long sentences in the article that might make readers feel somewhat overwhelmed, so it could be considered to break them into shorter sentences to enhance readability. Overall, the language quality of the manuscript is good, but there are still areas for improvement, particularly in the discussion and sentence structure, to ensure the article is more easily comprehensible and engaging to readers.

Reviewer 4 Report

Comments and Suggestions for Authors

This paper evaluates the reliability of the Smartpill for intra abdominal pressure. The authors placed the Smartpill in the stomach to pigs, and placed a wired intragastric sensor and a wired introperitoneal sensor as a reference. The data obtained from the three sensors were compared by statistical Spearman and Bland Altmann analysis. This study has a certain reference value for non-invasive detection of abdominal pressure. I think this paper can be published in the Sensors after major revision. Some comments are listed below.

1. The highest abdominal pressure in the experiment was only 15 mmHg (line 102), but the intraperitoneal pressure may be higher in critically patients. If the measurement can be made at a higher intraperitoneal pressure, the experimental results will be more convincing.

2. The data of only two pigs were collected in the experiment, which is difficult to ensure the accuracy and reliability of the study.

3. In this study, only experimental data for a short period of time were collected (line 139). In order to consider more situations and collect more data, longer monitoring is necessary.

4. Why is the measurement deviation very large when the abdominal pressure of two pigs is lower than 3mmHg (line 173 and line 202)?

5. If the pressure deviation of SmartPill measured in vivo is due to temperature, can the absolute pressure be obtained by appropriate temperature compensation?

6. Please add a photo and composition diagram of the Smartpill, and give it more introduction.

Comments on the Quality of English Language

None

Round 2

Reviewer 1 Report

Comments and Suggestions for Authors

The Authors revised sufficiently the mansucript. In my opinion the paper can now accepted for publication.

Reviewer 4 Report

Comments and Suggestions for Authors

The author carefully revised and explained the comments of the previous review. The work in this paper is more complete, so I suggest that it be published in Sensors journal. It is expected that the author's subsequent work will continue to supplement and improve this paper.

Comments on the Quality of English Language

None